# In Situ Monitoring of Kefir Fermentation Process Using Signal-Separable NMR Techniques

**DOI:** 10.3390/foods14061025

**Published:** 2025-03-18

**Authors:** Xiaoqi Shi, Yaoping Gou, Mengjie Qiu, Wen Zhu, Yanqin Lin

**Affiliations:** Fujian Provincial Key Laboratory of Plasma and Magnetic Resonance, Department of Electronic Science, Xiamen University, No. 422, Siming South Road, Siming District, Xiamen 361005, China; 33320200155802@stu.xmu.edu.cn (X.S.); nomankind2@gmail.com (Y.G.); qmj2874275229@163.com (M.Q.); 33320240156452@stu.xmu.edu.cn (W.Z.)

**Keywords:** fermented milk, nuclear magnetic resonance (NMR), in situ detection, pure shift, selective excitation

## Abstract

The fermentation time of fermented milk significantly influences its taste and nutritional value. Monitoring the fermentation process is crucial for ensuring the quality, flavor, and safety of fermented products. In this paper, the kefir fermentation process, as an example, was monitored in situ using advanced nuclear magnetic resonance (NMR) techniques. The fermentation process was tracked by obtaining pure shift spectra through the Pure Shift Yielded by Chirp Excitation (PSYCHE) sequence to separate heavily overlapped peaks, which enabled the identification and quantification of protons. The Gradient-Enhanced, Multiplet-Selective, Targeted-Observation NMR Experiment (GEMSTONE) sequence was employed to selectively excite the protons of interest in the overlapping region, enabling the rapid tracking of changes in the lactose and ethanol concentrations during fermentation. The results from these advanced NMR methods provide valuable insights into the dynamics of the kefir fermentation process, offering a more accurate and efficient way to monitor and control the fermentation of milk.

## 1. Introduction

The fermentation process of milk via kefir grains is a complex reaction process. The microorganisms in kefir grains are mainly composed of lactic acid bacteria, yeast, and a small amount of acetic acid bacteria [1]. Initially, lactic acid bacteria are activated by yeast and acetic acid bacteria to act on lactose, which is hydrolyzed by microbial lactase into the monosaccharides glucose and galactose. The lactic acid bacteria subsequently convert these monosaccharides into lactic acid, creating an environment conducive to yeast growth. This metabolic activity also produces ethanol, carbon dioxide, and various other compounds, contributing to the desirable taste and nutritional value of fermented milk [2]. Additionally, the macromolecular components of milk, such as proteins, are broken down in small amounts via bacterial catabolism into free amino acids, which provide a nitrogen source for lactic acid bacteria [3]. Fat plays a negligible role in the fermentation process. However, it can decompose into glycerol and fatty acids under acidic conditions, influencing the milk’s texture [4]. Figure 1 illustrates the kefir fermentation process. The primary change involves converting lactose into glucose and galactose, both of which are then metabolized into lactic acid, ethanol, and carbon dioxide. Ethanol is a distinctive byproduct of kefir-fermented milk, contributing to the unique flavor and effervescence of the fermented milk [5]. However, excessive ethanol can impair the taste and harm health. Therefore, the duration of fermentation is critical, as it influences the degree of fermentation, the residual lactose content, and the composition of the fermented milk, thereby affecting the taste and nutritional value of the final product [6]. By monitoring the fermentation process, the content of lactose remaining and ethanol produced can be assessed, allowing for the cultivation of fermented milk with varying lactose contents and tastes to meet diverse consumer needs.

There are various methods reported for monitoring the milk fermentation process, such as the mass spectrometry approach [7], microbial potentiometric sensors method [8], and multiple light scattering measurements [9]. However, most of these methods perform only qualitative or relative quantitative analysis and fail to provide accurate quantitative analysis of metabolites. Nuclear magnetic resonance (NMR) spectroscopy is an effective tool for the structural analysis and quantification of organic compounds and mixtures, enabling the nondestructive detection of metabolites [10], drug molecules [11], and biological molecules [12]. Initially developed for material structure identification and analysis, NMR technology has advanced to encompass applications such as elucidating chemical reaction mechanisms and monitoring chemical changes during reactions [13,14]. This non-invasive characterization method provides on-site information in real time, effectively preserving the original information of test samples. Other characterization techniques, such as infrared and Raman spectroscopy, require the extraction of specific substances [15,16], which can alter the milk’s structure, leading to the denaturation of proteins and polysaccharides, thus resulting in inaccurate detection and deviations in understanding the reaction process and mechanism [17]. Compared with these methods, NMR offers the distinct advantage of in situ reaction monitoring without causing damaging to the sample. Taking advantage of the features of NMR in situ monitoring, a series of 1D ^1^H spectra can be acquired during the reaction, making NMR especially useful for monitoring processes such as kefir fermentation. However, 1D measurements often suffer from insufficient resolution in complex mixtures due to overlapping peaks caused by the diversity of components and intricate coupling networks, which hinders the identification and quantification of protons, as seen in the spectrum of milk [18,19]. Increasing the spectral dimensions distributes information across multiple dimensions, thereby mitigating the issue of peak overlap [20]. The increase in dimensionality in 2D spectroscopy will result in an exponential increase in experimental time. For reactions in progress, the increase in acquisition time reduces the instantaneousness of the results.

The complexity of the spectra and the overlapping of spectral peaks are primarily attributed to a narrow chemical shift range and scalar coupling splitting [21]. A pure shift experiment (PSYCHE) (Figure 2a) is a well-established NMR method that addresses this issue by simplifying the spectra through the elimination of signal splitting caused by *J*-coupling, ensuring that each resonance corresponds to only one signal [22,23]. It employs two opposing swept-frequency pulses with small flip angle (β) and weak magnetic field gradients, which are combined with a 180° hard pulse to promote active spin refocusing while leaving passive spins unaffected to achieve homonuclear decoupling. Combining the hard 180° pulse and gradients, the evolution of the active spin remains constant while the passive spins dephase. The combination of chirp pulses and gradients can make the zero-quantum signal experience different evolution times at various positions, leading to the accumulation of different phases that eventually cancel out. The PSYCHE method offers excellent performance with higher sensitivity, spectral purity, and tolerance to strong coupling than previous broadband homonuclear decoupling methods, thus promising a wide range of applications.

For certain reactions, the quantitative information of a single substance is often sufficient, without requiring the entire spectrum. Thus, only the relevant peaks need to be selectively extracted, which simplifies the spectra and saves experimental time [24]. The Gradient-Enhanced, Multiplet-Selective, Targeted-Observation NMR Experiment (GEMSTONE), as shown in Figure 2b, selectively excites a single multiplet from a region of heavily overlapping proton signals in a single scan [25]. It uses both field gradients and chirp pulses for continuous spatial encoding of chemical shifts, preserving the on-resonance spin and dephasing the rest. The selective 180° pulse inverts only the spin of interest rather than the spins coupled to it. GEMSTONE exhibits great advantages regarding signal selectivity and resolution in a single scan, with the acquisition time close to that of 1D ^1^H method. It has great application potential in the detection of individual components in complex mixtures and reaction monitoring via NMR.

In this work, the PSYCHE and GEMSTONE NMR techniques were used to investigate the whole process of kefir fermentation. The pure shift spectra during the fermentation process were obtained using the PSYCHE method, converting the coupled multiplets into singlets. It simplifies the spectra, thereby facilitating the identification and quantification of individual peaks in milk and the fermentation product. GEMSTONE was used to track changes in lactose and ethanol during fermentation, which addresses the issue of significant peak overlap in conventional 1D ^1^H spectra. Compared to 2D spectra, it is more time efficient and provides clearer results. The quantitative data obtained can be used not only to monitor products during the fermentation process in real time but also to help optimize fermentation parameters, such as temperature, time, and strain selection. This can help optimize the quality of fermented dairy products, contributing to healthier, safer, and more nutritious options for human health.

## 2. Experimental Section

### 2.1. Sample Preparation

The milk used for testing was Anchor Whole Milk from New Zealand Milk Brands Limited (Auckland, New Zealand). It has a protein and fat content of 6% and a carbohydrate content of 2%. Kefir grain was purchased from a store named “YOGURT MAN” on the Alibaba platform. It is a kind of traditional leavening agent with an irregular shape and a curly, bumpy, or highly twisted surface, giving it a predominantly white or pale yellow color. The main components of kefir grains are water, sticky polysaccharides, proteolipids, and some probiotic bacteria inhabiting the grains, such as *Lactococcus*, *Lactobacillus*, and *Saccharomyces* [26]. These probiotics and other symbiotic organisms give kefir flora nutritional and health benefits. The kefir bacteria used were grown in pure milk after purchasing strains online. Deuterium oxide (D_2_O, 99.9% D) was provided by BY-BASF. All chemicals were not further purified.

The bacteria were first activated. The activation procedure followed manufacturer’s guidelines for kefir grain. The purchased dried kefir pellets were placed in a sterile glass bottle, and a glass of cool whole milk was added. The bottle was covered with gauze and secured with a rubber band. It was left at room temperature (25 °C) for 24 h without protection from light. This method was repeated for 3 to 7 days until the milk became thick, indicating that the activation was successful.

The sample consisted of 400 μL milk, 200 μL of D_2_O, and 200 μg of kefir granules. They were placed together into a nuclear magnetic tube. Sampling started after standing for 60 min. The fermentation process of the milk was tested over the subsequent 12 h to observe changes in its composition. To minimize random errors and enhance the accuracy and reliability of the results, the fermentation experiment was repeated three times under identical conditions for testing, and the averaged results were used.

### 2.2. NMR Experiments

All NMR experiments were performed on a Varian NMR spectrometer of 11.7 T (500 MHz proton resonance frequency, Agilent Technologies Inc., Santa Clara, CA, USA). The instrument has a 5 mm H/C probe, a 54 mm narrow cavity, and a Z-direction pulse gradient with a maximum intensity of 60 G/cm. The experimental temperature was maintained at room temperature (298 K). The experimental data were processed using VnmrJ 4.2, provided by the supplier, and MATLAB 2020.

The pulse sequence of PSYCHE is shown in Figure 2a. It uses low flip angle (β) swept pulses in the presence of weak magnetic field gradients. It is a pseudo-two-dimensional experiment implemented by splicing each incremental data block. Since the new Free Induction Decay (FID) value is reconstructed from the original data blocks, the spectral resolution increases with the number of spliced blocks. The 90° pulse duration measured was 10.6 μs for ^1^H. The spectral width was 5000 Hz (10 ppm). The carrier frequency was set to 5 ppm. The duration, pulse shape, and flip angle of the adiabatic pulses in the PSYCHE sequence were then set. The chirp pulse shape of ‘Wurst’ was used. The system then automatically calculated the RF power of the chirp pulse element based on the bandwidth, flip angle, and the excitation pulse duration and power of the 90° pulse. The desired coherent transfer paths were selected using G_1_ and G_2_ gradients with amplitudes of 24 G cm^−1^ and 38 G cm^−1^ and a duration of 1.5 ms, respectively. G_3_ acted as a spatial encoder with an intensity of 0.8 G cm^−1^. The chirp pulses had a 20° flip angle, a power of 7 dB, a duration of 15 ms, and a scanning width of 10,000 Hz. The parameters were set with recovery delay d_1_ = 1 s and scan number nt = 4, and the pre-saturation water suppression module was activated. The FID was 0.8 s, with 40 blocks stitched together. Sampling occurred every 0.5 h, with a collection time of 5 min, and the results were compared with 1D ^1^H spectra.

The GEMSTONE sequence is shown in Figure 2b. The waveform ‘Wurst’ of the chirp pulse is also used, which provides selectivity of the sequence. It has a scan width of only 2500 Hz because it only excites the interested signal rather than the entire spectrum. A selective 180° pulse, with a bandwidth of 100 Hz and a duration of 18.5 ms, is employed with an ‘RSNOB’ shape pulse to refocus *J* modulation. The gradient pulses are desired to select a coherence transfer pathway without phase cycling. They have amplitudes of 9 G cm^−1^ and duration of 1.5 ms. The parameters were set with recovery delay d_1_ = 1 s and scan number nt = 4, and the pre-saturation water suppression module was activated. Sampling occurred every 0.5 h, with a collection time of 30 s.

## 3. Results and Discussion

Figure 3 shows the traditional 1D ^1^H NMR spectrum of milk, where the water signal at about 4.8 ppm was significantly suppressed. At the low-frequency area of 0.5~2.0 ppm, the acyl chain signals mainly belong to the fat of milk (Figure 3a). The resonance signals between 2.2 and 3.0 ppm are attributed to trace compounds in milk, and the characteristic peaks of the methyl group protons in acetate appear at 2.3 to 2.5 ppm. The proton from the methyl group in creatinine is observed at 2.85 ppm. The few remaining peaks with broadening signals are caused by unsaturated fatty acids in milk fat, as shown in Figure 3b. The signal around 3.0 ppm is the spectral peak of the proton in the methyl group attached to the N atom in lecithin, and the weak peaks of lecithin concentrate at 4.0–4.5 ppm. Lactose signals concentrate between 3.0 and 4.0 ppm, primarily involving protons on sugar rings and methylene groups stacked and overlapping, making accurate identification challenging. Protons in the hydroxyl groups are active hydrogen atoms that can be chemically exchanged with the protons in the water, resulting in a severe broadening of the line width, with chemical shifts at 6.5–8.5 ppm, being difficult to observe in the spectra [27]. Due to the high molecular weight and low molar concentration of proteins, pre-saturation water suppression results in reduced intensities of these broad signals [28]. Therefore, the peaks in the region of protein are extremely weak and overlapping, even almost invisible, as shown in Figure 3d. Comparing the signals produced by milk fat, the lactose signals are narrower and more sensitive, while the signal for milk fat is broad (Figure 3b,c). This is because lactose is more soluble in milk, whereas milk fat has more acyl chains that keep it suspended in the emulsion [29]. Although the traditional 1D ^1^H spectrum can provide a rough characterization of milk substances, the complexity of milk’s composition and the overlapping peaks in the sugar region present significant challenges for precise identification and quantification. Particularly during the kefir fermentation process, a primary chemical change is the decomposition of lactose, which cannot be effectively monitored using 1D ^1^H NMR spectroscopy.

Therefore, the advanced NMR technique PSYCHE was used to obtain the pure shift spectrum of milk. As shown in Figure 4, the individual components were labeled using different colors. Compared with the traditional 1D ^1^H method, PSYCHE reduces the spectral overlap and greatly increases the spectral resolution by suppressing *J*-coupling. Especially in the lactose region of 3.4–3.8 ppm, the protons on the sugar rings are effectively distinguished, which simplifies the assignment of the milk spectrum (Figure 4a, expanded area). The specific peak positions are assigned as shown in Table 1. However, using small flip angles in adiabatic pulses reduces the spectral sensitivity, causing the signals of trace elements to be obscured by noise. The signal-to-noise ratio (SNR) of the PSYCHE spectra was only 12% of that of the 1D ^1^H spectra, but it did not affect the quantification of the primary substances undergoing changes during fermentation.

At the beginning of fermentation, lactose is partially hydrolyzed into glucose and galactose under the action of lactic acid bacteria. Glucose is metabolized via the Embden–Meyerhof–Parnas (EMP) pathway, where it is converted into pyruvate. Subsequently, pyruvate enters three metabolic pathways. The first metabolic pathway is the further reduction of pyruvate to lactic acid by lactic acid bacteria. At the same time, pyruvate under anaerobic conditions undergoes alcoholic fermentation to produce ethanol. In addition, pyruvate undergoes decarboxylation to form acetyl-CoA, which enters the tricarboxylic acid cycle (TCA cycle). Within the TCA cycle, acetyl-CoA is further oxidized to citric acid and subsequently to carbon dioxide and water. Meanwhile, galactose is phosphorylated by galactokinase and enters the glucose metabolism pathway to also be broken down [30]. This intricate conversion process is better understood as a metabolic chain rather than a simple equation. Ultimately, the fermented milk is in a semi-solid form with poor fluidity. As shown in Figure 4b, the metabolites of milk after 6 h of fermentation were analyzed using the PSYCHE sequence. During the initial stage of fermentation, lactic acid bacteria rapidly multiply, enhancing lactic acid production. The characteristic peak of the methyl group in lactic acid was observed at 1.21 ppm, despite partial overlap with the lecithin resonance peak at 1.20 ppm. The production of lactic acid lowers the pH of the fermentation broth, creating a favorable environment for yeast. Also, it ensures the safety and stability of the fermentation process by inhibiting the growth of harmful microorganisms. In addition, lactic acid accumulation inhibits further growth of lactic acid bacteria, while yeast continues to break down sugars to produce ethanol. The evidence of ethanol produced during fermentation was demonstrated at 1.02 ppm. In the carbohydrate metabolism region (3.0–4.0 ppm), some characteristic peaks of incompletely metabolized glucose were observed, likely due to the short fermentation time, preventing full conversion to lactic acid and other metabolites. Furthermore, citric acid was not detected in the 2.4–2.6 ppm region, likely because it is rapidly consumed and converted into carbon dioxide and water or further utilized in cellular metabolism as an intermediate product.

The kefir fermentation process was characterized using 1D ^1^H spectra and PSYCHE spectra to monitor the chemical changes, respectively. The 1D ^1^H NMR spectra reflect the overall trend of metabolite changes during fermentation. However, severe signal overlap, especially in the 3.4–3.8 ppm range, hindered the accurate quantification of specific compounds due to interference from resonance peaks, as shown in Figure 5a. In contrast, the PSYCHE sequence enhances spectral resolution by retaining chemical shift information while removing coupling information, as shown in Figure 5b. It effectively separates signals in the crowded region, thus benefiting the quantification of metabolites. The signal at 1.47 ppm of CH_2_ in the fatty acid acyl chain was used as an external quantitative reference, which does not participate in the fermentation reaction. As shown in Figure 5c, the concentration of lactose, indicated by the change in CH-4′ at 3.82 ppm, gradually decreased from 35.6 mM to 26.5 mM over 12 h, with the consumption rate decreasing over time. Lactic acid production commenced only after 6 h of fermentation. Despite the PSYCHE sequence significantly enhancing the spectral resolution, accurately extracting the characteristic methyl peak of lactic acid remained challenging due to the broad resonance peak of lecithin near 1.21 ppm. Since lecithin does not change during fermentation, the production of lactic acid can be indirectly assessed by quantifying the integral value of the overall signal at 1.20 ppm. Meanwhile, the PSYCHE sequence successfully separated the characteristic peak of methyl in ethanol at 1.02 ppm. The change in its concentration went through three stages: a slow increase during the initial phase of fermentation (0–3 h), a rapid generation phase lasting from 3 to 6 h, and a stabilized level of about 3 mM after 6 h. Additionally, a weak citric acid signal was detected in the 2.4–2.6 ppm range only during the initial phase of the reaction, which disappeared rapidly as fermentation progressed. This observation suggests that citric acid quickly enters the TCA cycle and is metabolically consumed during fermentation. Using the PSYCHE sequence to characterize kefir fermentation greatly simplified the spectra compared to traditional 1D ^1^H NMR spectra, effectively separating severely overlapping peaks and facilitating spectral analysis and quantification. Although two-dimensional spectra can also separate overlapping peaks, they require longer acquisition times for good resolution. In contrast, PSYCHE shortens the experimental time and can better reflects the instantaneous state of the reaction.

The GEMSTONE sequence selectively excited the signals for lactose and ethanol at 3.82 ppm and 1.02 ppm, respectively, which allowed for the quantification of changes in these peaks throughout the reaction, as shown in Figure 6. The spectra contain information on only the peaks of interest and retain the coupling information. Figure 6a shows the peak at 3.82 ppm for the proton of CH-4′ in lactose, which forms a triplet due to coupling with the neighboring protons on the sugar ring. As the fermentation process progresses, a significant decrease in the signal intensity was observed. Figure 6c shows the quantification results of the intensities of CH-4′ using the GEMSTONE spectra. During the initial 9 h of fermentation, the quantification results obtained using the GEMSTONE and PSYCHE methods exhibit high consistency, indicating that the GEMSTONE spectra provide comparable quantitative accuracy to the PSYCHE spectra in the early stages of fermentation. However, the GEMSTONE spectra demonstrated a faster decline in lactose concentration 9 h later. The content of ethanol was quantified according to the spectra in Figure 6b. Ethanol increased slowly in the first 3 h, and the ethanol production rate increased around 3–6 h. Subsequently, the increase in ethanol content leveled off, consistent with the results obtained from the PSYCHE spectra. However, the signal intensity increased first and then slightly decreased after 9 h, as shown in Figure 6d. This is because as fermentation advances into the later stages, the fermented milk sample gradually solidifies and produces bubbles, significantly reducing sample homogeneity, which has a noticeable effect on the GEMSTONE spectra. While the PSYCHE spectrum is also affected by sample heterogeneity, it includes stable reference peaks that allow for effective normalization to correct changes in signal intensity. In contrast, the GEMSTONE spectrum contains only the characteristic peaks of the target metabolites and lacks internal reference standards, making it incapable of accurately quantifying the metabolite content when the sample causes magnetic field inhomogeneity. Figure 7 summarizes the trends in the concentrations of lactose and ethanol obtained using the two methods during the fermentation process. The PSYCHE method demonstrated more excellent quantitative reliability in the case of significant changes in the physical state of the samples during the later stages of fermentation. However, PSYCHE spectra have a lower signal-to-noise ratio (SNR) due to its utilization of a small flip angle, which achieves only 5–20% of the sensitivity of 1D ^1^H spectra. In contrast, GEMSTONE spectra can achieve 80% of the sensitivity of 1D ^1^H spectra. In addition, the method does not require phase cycling, and only a single scan is needed to obtain a high-resolution spectrum. These features make the GEMSTONE method more suitable for monitoring fast reactions in uniform magnetic fields.

The PSYCHE and GEMSTONE techniques provide the ability to monitor the fermentation process in real time, offering a rapid and efficient way to track changes in key components such as lactose and ethanol concentrations. Compared to traditional 2D NMR methods, PSYCHE and GEMSTONE offer shorter acquisition times, which can be beneficial in high-throughput industrial environments. This allows for more frequent acquiring and quicker decision making regarding process adjustments, reducing the production time and improving the overall efficiency. In summary, integrating these advanced NMR techniques into QC procedures in industrial fermented milk manufacturing could enhance product consistency, safety, and quality by providing more precise, real-time, and non-destructive monitoring of the fermentation process.

## 4. Conclusions

In this study, the pure shift spectra and selective excitation spectra of kefir fermented milk were successfully obtained by using the advanced PSYCHE sequence and GEMTSTONE sequence, which realized the real-time monitoring and quantitative analysis of the specific proton signal changes during kefir fermentation. The methods enable the in situ exploration of the fermentation process of kefir milk and indirectly reflect the kinetic characteristics of the fermentation reaction by tracking the signal changes of specific protons in lactose and ethanol. Compared with conventional 1D ^1^H spectra, the methods significantly improve signal extraction and quantification difficulties by effectively resolving the severe overlapping of spectral peaks in complex mixtures. Meanwhile, compared with the 2D NMR technique, the spectra obtained using these methods are relatively simple, and the acquisition time is significantly shortened, which improves the time resolution and is more suitable for the real-time monitoring of dynamic processes. The non-destructive in situ characterization method based on the PSYCHE and GEMSTONE sequences provides new technical means for the dynamic monitoring of chemical reaction processes and demonstrates the broad application prospects of dynamic NMR technology in the research fields of food fermentation and biometabolism.

## Figures and Tables

**Figure 1 foods-14-01025-f001:**
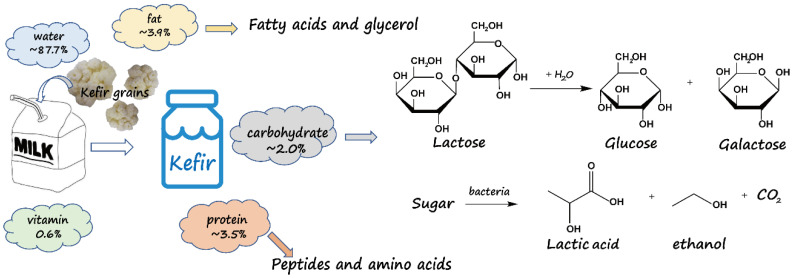
The whole process of kefir fermentation and the change of lactose.

**Figure 2 foods-14-01025-f002:**
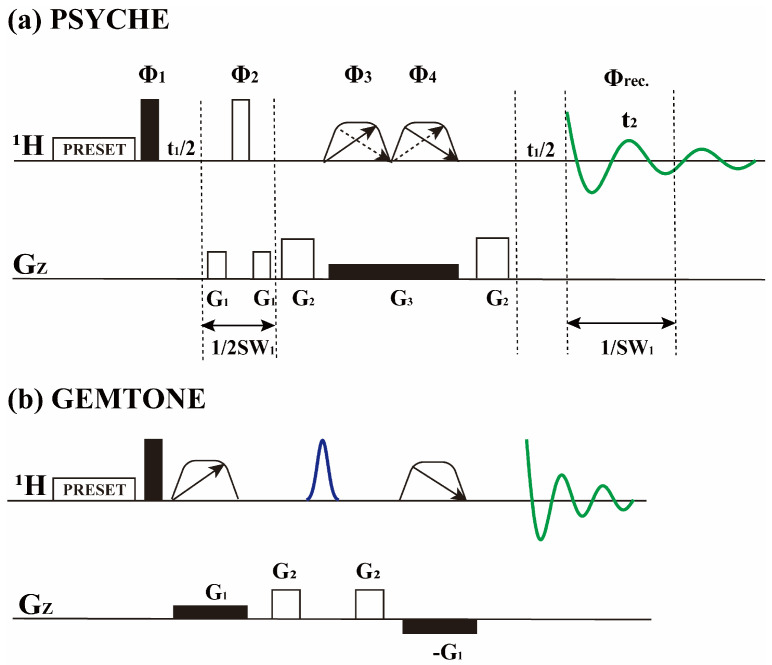
(**a**) PSYCHE pulse sequence. It is a pseudo-2D experiment, splicing to obtain a 1D spectrum. (**b**) GEMSTONE pulse sequences. Solid and hollow rectangles denote 90° and 180° pulses, respectively. The trapezoid shape indicates the chirp pulse, where the arrow indicates the sweep direction. The sinc shape represents a band-selective pulse. The chirp pulses are applied accompanied with the weak gradients G_3_ in PSYCHE and G_1_ in GEMSTONE, respectively. The gradients G_1_ and G_2_ in PSYCHE and G_2_ in GEMSTONE are the coherence selection gradients, respectively.

**Figure 3 foods-14-01025-f003:**
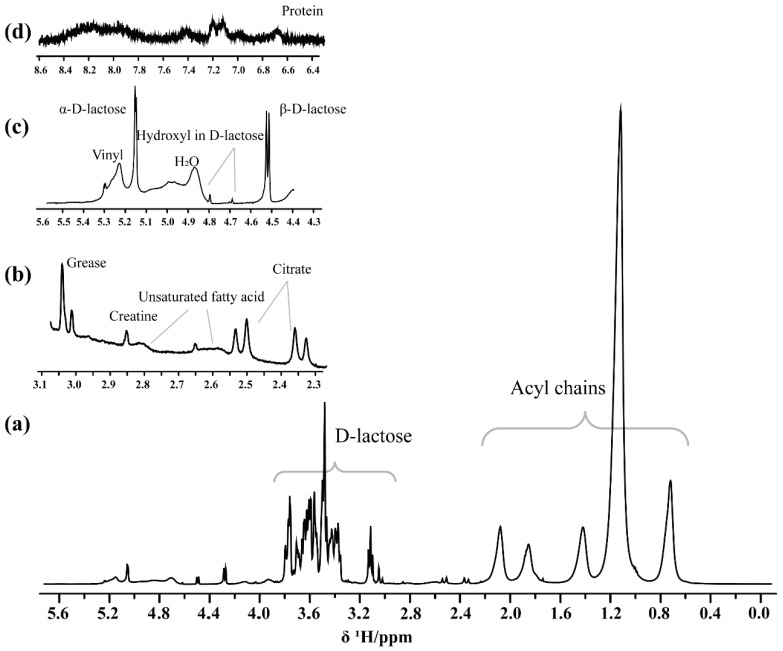
(**a**) Traditional 1D ^1^H NMR spectrum of milk acquired on a 500 MHz spectrometer. (**b**) The magnified region of the protein, with signals corresponding to amide and aromatic protons. (**c**) The proton signals of α-D-lactose and β-D-lactose. (**d**) The presence of trace metabolites such as grease, unsaturated fatty acids, citrate, and creatine in the 2.3 to 3.1 ppm range.

**Figure 4 foods-14-01025-f004:**
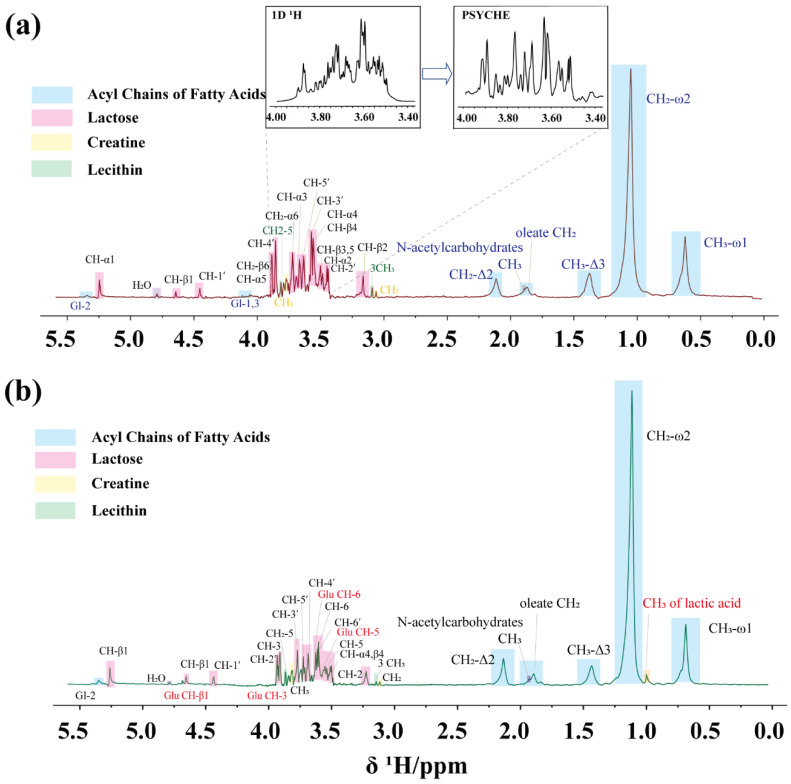
(**a**) The PSYCHE spectrum of milk. The expanded area in the figure compares the 1D ^1^H spectrum with the PSYCHE spectrum of lactose at 3.4–4.0 ppm. (**b**) The PSYCHE spectrum of milk after 6 h of fermentation, where the peaks of the newly generated substances are labeled in red text. The blue area shows signals for fatty acid acyl chains, the pink area shows signals for lactose, the yellow area shows signals for creatine, and the green area shows signals for lecithin.

**Figure 5 foods-14-01025-f005:**
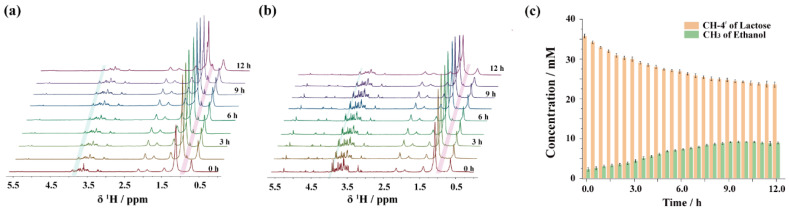
(**a**,**b**) The spectra acquired at 30 min intervals for 12 h during the fermentation process of milk. (**a**) 1D ^1^H spectra. The acquisition time for each spectrum was 2 s. (**b**) PSYCHE spectra. The acquisition time for each spectrum was 5 min. (**c**) The CH-4′ of lactose and the CH_3_ of ethanol produced in fermented milk during fermentation were quantified. (Figure 6a,b show labeled positions of the signals).

**Figure 6 foods-14-01025-f006:**
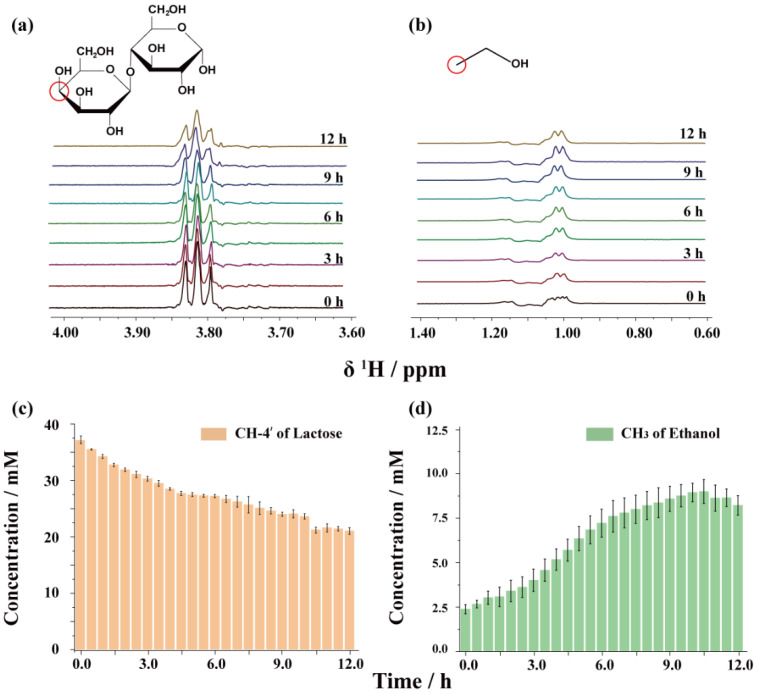
GEMSTONE spectra during kefir fermentation process. The samples were acquired every 0.5 h, and the experimental time for each spectrum was 40 s. (**a**) The changes in GEMSTONE selection of the signal CH-4′ of lactose at 3.82 ppm. (**b**) The changes in GEMSTONE selection of the signal -CH_3_ of ethanol at 1.02 ppm. (**c**) Variation curve of the integral area of CH-4′ of lactose characteristic peaks in fermentation for 12 h. (**d**) Variation curve of the integral area of -CH_3_ of ethanol characteristic peaks in fermentation for 12 h.

**Figure 7 foods-14-01025-f007:**
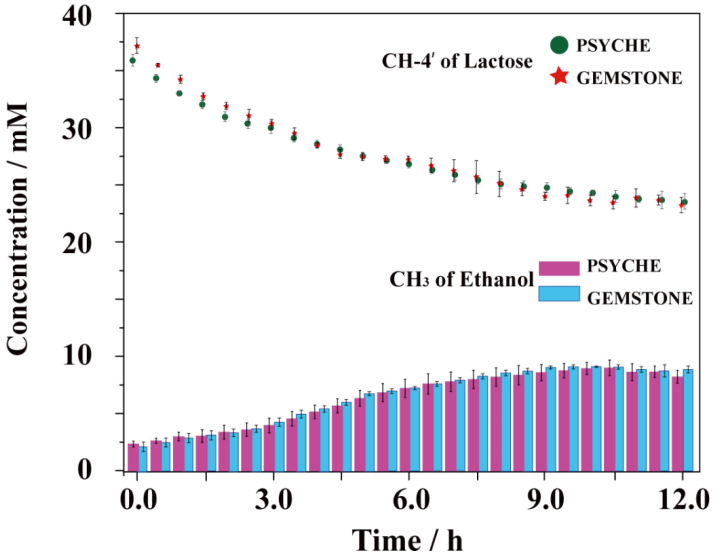
The trendlines for the changing concentrations of lactose and ethanol, as shown by both the PSYCHE and GEMSTONE methods.

**Table 1 foods-14-01025-t001:** Assignment of PSYCHE ^1^H NMR signals for lactose from milk.

Assignment	Chemical Shift (ppm)
CH-β2	3.15
CH-2′	3.47
CH-α2	3.47
CH-β5	3.49
CH-β3	3.50
CH-α4 β4	3.53
CH-3′	3.55
CH-α3	3.66
CH_2_-α6 β6	3.72
CH-4′	3.82
CH_2_-6′	3.84
CH-α5	3.84
CH-1′	4.39
CH-β	4.63
CH-α1	5.24

## Data Availability

The datasets used and analyzed during the current study are available from the corresponding author on reasonable request due to privacy.

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
