# Peer review of "In Situ Monitoring of Kefir Fermentation Process Using Signal-Separable NMR Techniques"

_foods, 2025, doi:10.3390/foods14061025_

Round 1

Reviewer 1 Report

Comments and Suggestions for Authors

All suggestions were included in the attached file.

Comments on the Quality of English Language

There are minor issues, but the text reads very well.

Author Response

Response to Reviewer 1 Comments

1. Summary

Thank you very much for taking the time to review this manuscript. We are grateful to the reviewers for stimulating and constructive comments. We have carefully revised the manuscript following these suggestions. Please find the detailed responses below and the corresponding revisions/corrections highlighted/in track changes in the re-submitted files. In the following, the original comments of the reviewers are in colored (blue), and our response follows each comment in black.

2. Questions for General Evaluation

Reviewer’s Evaluation

Response and Revisions

Does the introduction provide sufficient background and include all relevant references?

Yes/Can be improved/Must be improved/Not applicable

[Please give your response if necessary. Or you can also give your corresponding response in the point-by-point response letter. The same as below]

Is the research design appropriate?

Yes/Can be improved/Must be improved/Not applicable

Are the methods adequately described?

Yes/Can be improved/Must be improved/Not applicable

Are the results clearly presented?

Yes/Can be improved/Must be improved/Not applicable

Are the conclusions supported by the results?

Yes/Can be improved/Must be improved/Not applicable

3. Point-by-point response to Comments and Suggestions for Authors

Comments 1: General concept comments- This study provides insights into the performance of novel, NMR-based methodologies to measure the progression of milk fermentation, taking as an example the process of kefir making. The application of these methods in such a setting is a novel concept and hence it required a detailed demonstration of method performance, which was achieved in this paper. Authors have clearly shown the advantage of using PSYCHE and GEMSTONE over a standard 1D 1H NMR technique. These results offer valuable insights for individuals seeking to control or get detailed information on the milk fermentation process.

The reviewer suggests minor corrections to enhance the quality of the manuscript. All the current content is appropriate, however, the paper lacks discussion and crucial information in the Introduction about alternative methods for milk fermentation monitoring. In addition, some minor changes to figures and text should be considered to improve the readability of the whole manuscript.

The manuscript requires minor changes to the style and correcting some grammar issues. Despite this, the text in its current version communicates the message effectively.  

Response 1: Thanks for reviewer’s suggestions. We agree with this comment. Therefore, we have added information in the introduction alternative methods for milk fermentation monitoring at page 2 lines 6-10.

Comments 2: Abstract- Is brief and reads well, besides some stylistic issues, such as in the first sentence. I would recommend enriching the abstract with 1-2 sentences more on the importance of monitoring the progression of milk fermentation to underscore the impact of the paper. In addition, a mention on what type of milk fermentation was researched would be desirable.

Response 2: We agree with this comment. We have added information on the importance of monitoring the milk fermentation process based on your suggestions and emphasized the type of fermentation in the Abstract.

Comments 3:  Introduction- The introduction is opened by a well-written summary of kefir fermentation process. Figure 1, as an addition to this part, is of a very good quality and supports the read. However, please be consistent and use nomenclature such as kefir grains or kefir microbiota rather than kefir bacteria, since as mentioned, kefir grains contain not only bacteria but also yeast. In addition, please change the word “yogurt” to “kefir”. This comment should be applied through the text.

 As a reader, I would prefer Figure 1 and related text to follow with the information about the methods that are currently used to monitor the progression of milk fermentation. This is not mentioned in the entire paper, but seems important because the authors present a novel method that should be somehow compared with standard approaches to underscore the impact of the change it could make when applied.

 The description of a novel NMR method to monitor the milk fermentation progression is very appropriate, since the method is not used as a standard. The authors supported the read with a figure 2 which is also helpful.

Some other minor comments to the introduction part:

On page 3, a following wording was used “whole process of milk fermentation into yogurt by kefir”- I disagree with such phrase. Yoghurt is not kefir, please revise this part.

Please revise the last sentence in this section. A monitoring method cannot improve anything, it can only supply data.

Response 3: Thanks for reviewer’s suggestions. We apologize for the error.

First, we have fixed the error in Figure 1 and changed all “yoghurt” to “kefir”.

Then, we mentioned alternative methods, such as Infrared and Raman, on page 2, line 17, but this was not comprehensive. Therefore, we added a section at the beginning of the paragraph summarizing previously reported methods for monitoring the fermentation process and briefly outlining their drawbacks.

Finally, in the last sentence, we would like to express that the optimization of parameters can help to improve the quality of fermented dairy products. Based on your suggestion, we have revised in the manuscript (page 3 line 30) to avoid ambiguity.

Comments 4: Materials and methods- please address following comments:

 â–ª _At the beginning of page 4, please use italics for microbial genera designations.

                   â–ª Please also reference where the information about the contents of kefir grains comes from.

                   â–ª Grains were purchased online, but please inform about the supplier or the manufacturer.

                   â–ª Second paragraph of section 2.1- please give information whether the activation procedure was following kefir grain manufacturer’s guidelines or what other source was it from?

Response 4: Thanks for reviewer’s questions. We have corrected the mistakes in the Materials and methods section of the manuscript and added the details accordingly in response to the reviewer's questions.

The microbial genera designations have been italicized at page 4 line 9.

The supplier of grains was added to the Experiment Section at page 4 line 4.

The activation procedure was following kefir grain manufacturer’s guidelines (page4 line13).

Comments 5: Results are adequately described. Section is opened with information on where specific peaks appear on the spectra, which is referenced. This all reads well, but does not make a discussion. The article needs some discussion, which would be either focused on expected changes in microbial metabolites over time or performance of the methods that were used for a similar purpose. In addition, some minor grammar and style issues could be refined.

Response 5: Thanks for reviewer’s suggestions. We have added a discussion of the behavior of the microorganisms after 6 hours of kefir fermentation combined with PSYCHE spectra in the Results and Discussion section on page 8, line 13-27.

Comments 6: I would also recommend changes in the figures to support the read of the text better and inclusion of statistical analyses as described below:

Figure 3- please add designations a-d to each of the graphs and use them in the figure caption instead of top, second and third. The bottom graph is not described in the caption, but it should be. This comment is only valid if Figure 3 will not be formatted further.

Response 6: Thanks for reviewer’s suggestion. We have added designations (a-d) to Figure 3 and corrected in manuscript.

Comments 7: Figure 3 and 4 are compared within the text, but the difference is not so obvious to the reader. Since the lactose region of the results is of special interest, consider enlarging it and compare between methods on a single figure.

Response 7: Thanks for reviewer’s suggestion. We have enlarged the 1D 1H spectrum and the PSYCHE spectrum of the lactose region, and displayed them in Figure 4(a).

Comments 8: Figure 4 and 5- a spelling mistake is present on the figures where it should say Acyl. These two figures are also compared in the text and separated by it. Consider placing these two graphs on a single figure to facilitate a direct comparison to the reader or leave only one graph, since you have compared both on Figure 6.

Response 8: Thank you for pointing this out. We sincerely apologize for such a careless mistake and have corrected it in the manuscript. To facilitate direct comparison for the readers, we have merged the original Figures 4 and 5 into one figure, as shown in Figure 4(a) and (b)

Comments 9: Figure 6- Here authors set 1D 1H and PSYCHE spectra on one figure, which is a good decision and helps to compare both methods, but also multiplies the content unnecessarily. Consider removing figure 3 and, as suggested in the third comment to this section, enlarging the region for lactose in both methods, this would be of interest on its own since these peaks are relatively small due to comparatively large 2.25-0.5 ppm region. I also think that showing either Figure 4 or 5 would be of benefit to the paper, but not both, since the comparison of both graphs is already visible on Figure 6. This should trigger adequate changes in the text. First, the regions where exact peaks from NMR and PSYCHE were appearing should be described, and then, the comparison of both should be performed supported with the content of Figure 6 and the current figure 3 and 4 combined.

Response 9: Thanks for reviewer’s suggestion. To avoid repetition, we have deleted the original Figure 6(c) and made adjustments to Figures 3, 4, and 5, including the addition of an enlarged view of the lactose region in Figure 4(a), which more clearly demonstrates the advantages of the PSYCHE method.

Comments 10: Concerning the following text “During the initial 9 h of fermentation, the quantification results obtained by GEMSTONE and PSY-CHE methods exhibit high consistency, indicating that the GEMSTONE spectra provide comparable quantitative accuracy to the PSYCHE spectra in the early stages of fermentation. However, GEMSTONE spectra demonstrated a faster decline in lactose concentration 9 h later.” It would be of benefit to compare two results on a single graph. Consider making a single figure with trendlines for changing concentration of lactose and ethanol as shown by both methods. In addition, use statistics to compare whether these results are statistically significantly different. There are error bars on the columns so I suspect that the results were means of several replicates. Please describe what you have done in the method section, how many replicates, what measure of central tendency and data spread were used. This information is also suitable to include in the figure captions.

Response 10: Thanks for reviewer’s suggestion. We have made a single figure with trendlines for changing concentration of lactose and ethanol as shown by both PSYCHE and GEMSTONE methods, as shown in Figure 7 in manuscript. The calculated p>0.05, indicating that these results are no statistically significantly different. In addition, the number of repetitions of the experiment has been described in the Methods section.

Comments 11: Conclusions- Are well written and evidence-based. This section should be modified and enriched with new insights once the authors add the discussion to the Results and Discussion section.

Response 11: Thanks for reviewer’s suggestion. We have added content to the Results and Discussion section.

Comments 12: Figures and Tables- Figures (including graphical abstract) and tables are of good quality. It seems like the authors have made an effort to prepare data presentation. The result is pleasant to the eye. However, I have some comments to address to reflect comparisons that are made in the text better. These were all included in the comments above. And one more comment concerning graphical abstract, you have made kefir not yogurt, please change the description of your end product and include the word “grains” next to “kefir” that is currently present on the figure.

Response 12: Thank you for the suggestions. We have revised the graphical abstract according to the suggestions.

Comments 13:  References- The list contains 26 references of which more than a half is older than 2020. Nevertheless, all references are appropriately used. However, for a full scientific paper their number is quite low. I believe the authors could enrich the text with more relevant and new references if they would decide to include a discussion and some information about other methods used for monitoring the fermentation process, which would be of benefit to this work.

Response 13: Thank you for the suggestions. We have added the content regarding the points you mentioned and references (7-9, 26) in the manuscript.

4. Response to Comments on the Quality of English Language

Point 1: There are minor issues, but the text reads very well.

Response 1: Thank you for the constructive comments. We carefully checked for language issues and corrected them.

5. Additional clarifications

Reviewer 2 Report

Comments and Suggestions for Authors

The present manuscript demonstrates the application of PSYCHE and  GEMSTONE techniques to enhance the effectiveness of ¹H-NMR measurements in milk kefir fermentation. The increased spectral resolution, improved NMR signal extraction and quantification of metabolites formed during kefir fermentation were investigated. The manuscript was adequately prepared but a more comprehensive version with an extensive discussion should be amended. In my humble opinion, several points as outlined below should be considerably clarified and revised.

  • Title: To be more specific to the context of the study, I suggest using the term “kefir fermentation” instead of “milk fermentation.”
  • Abstract: The abstract should be re-checked again after considering the collective comments below. Please clarify the term 'fermentation degree.' Be specific about whether this study focuses on 'kefir' fermentation. The last sentence should clearly indicate whether it refers to controlling the 'milk fermentation process' or the 'kefir fermentation process.
  • There is no Line No. indication in the reding version of manuscript.
  • Figure 1: Inoculating kefir grains into a liquid milk base does not produce a yogurt product. Please refer to the FAO/WHO Codex Standard for Fermented Milks for the definitions of different types of fermented milk.
  • Introduction paragraph 2: The NMR metabolomics also has certain limitations in detecting and measuring aroma or volatile metabolites produced during food fermentation. Please reconsider.
  • Introduction paragraph 5: Inoculating kefir grains into a liquid milk base does not produce a yogurt product. Please refer to the FAO/WHO Codex Standard for Fermented Milks for the definitions of different types of fermented milk.
  • Section 2.1, Paragraph 1: Please specify the brand or manufacturer of the kefir grains used in this study. If artisanal cultures were used, kindly provide the source information as well.
  • Section 2.1, Paragraph 2: I recommend changing the fermentation temperature into degree Celsius.
  • Section 2.1, Paragraph 3: How can the authors ensure that the micro-fermentation (in-vial) performed in this study was reliable, applicable, and produced results comparable to traditional kefir fermentation?
  • Figure 3, 4 and 5: It would be beneficial for the authors to present a combined figure with overlaid spectra, highlighting specific regions to demonstrate the effectiveness of PSYCHE and GEMSTONE techniques in enhancing resolution, improving signal extraction, and enabling better quantification compared to traditional 1H-NMR spectra.
  • Results and discussion: In the last paragraph, the potential applications and benefits of the improved NMR techniques by PSYCHE and GEMSTONE investigated in this study, particularly in relation to the QC procedures in industrial fermented milk manufacturing, should be suggested.
  • Conclusions: The conclusion should be re-checked again after considering the collective comments above.

Author Response

Response to Reviewer 2 Comments

1. Summary

Thank you very much for taking the time to review this manuscript. We are grateful to the reviewers for stimulating and constructive comments. We have carefully revised the manuscript following these suggestions. Please find the detailed responses below and the corresponding revisions/corrections highlighted/in track changes in the re-submitted files.

2. Questions for General Evaluation

Reviewer’s Evaluation

Response and Revisions

Does the introduction provide sufficient background and include all relevant references?

Yes/Can be improved/Must be improved/Not applicable

Is the research design appropriate?

Yes/Can be improved/Must be improved/Not applicable

Are the methods adequately described?

Yes/Can be improved/Must be improved/Not applicable

Are the results clearly presented?

Yes/Can be improved/Must be improved/Not applicable

Are the conclusions supported by the results?

Yes/Can be improved/Must be improved/Not applicable

3. Point-by-point response to Comments and Suggestions for Authors

Comments 1: Title: To be more specific to the context of the study, I suggest using the term “kefir fermentation” instead of “milk fermentation.”

Response 1: Thank you for pointing this out. We agree with this comment. Therefore, we have changed the title to “In situ monitoring of kefir fermentation process by signal-separable NMR techniques.”

Comments 2: Abstract: The abstract should be re-checked again after considering the collective comments below. Please clarify the term 'fermentation degree.' Be specific about whether this study focuses on 'kefir' fermentation. The last sentence should clearly indicate whether it refers to controlling the 'milk fermentation process' or the 'kefir fermentation process.

Response 2: Thank you for pointing this out. Accordingly, we have emphasized the fermentation process as kefir fermentation. The manuscript focuses on the effect of fermentation time on the material composition of fermented milk, thus we have changed the “fermentation degree” to “fermentation time”.

Comments 3: There is no Line No. indication in the reding version of manuscript.

Response 3: Thanks for reviewer’s suggestions. We have added line numbers to the manuscripts.

Comments 4: Figure 1: Inoculating kefir grains into a liquid milk base does not produce a yogurt product. Please refer to the FAO/WHO Codex Standard for Fermented Milks for the definitions of different types of fermented milk.

Response 4: Thanks for reviewer’s suggestions. We have modified Figure 1 refer to the FAO/WHO Codex Standard for Fermented Milks.

Comments 5: Introduction paragraph 2: The NMR metabolomics also has certain limitations in detecting and measuring aroma or volatile metabolites produced during food fermentation. Please reconsider.

Response 5: We agree with the reviewer. The NMR method does have limitations in detecting and measuring aromatic or volatile metabolites. Therefore, we have removed the description of the comparison of measuring aroma or volatile metabolites produced during food fermentation from the manuscript.

Comments 6: Introduction paragraph 5: Inoculating kefir grains into a liquid milk base does not produce a yogurt product. Please refer to the FAO/WHO Codex Standard for Fermented Milks for the definitions of different types of fermented milk.

Response 6: Thanks for reviewer’s suggestions. We've changed all “yogurt” to “kefir” or “fermented milk” according to the FAO/WHO Codex Standard.

Comments 7: Section 2.1, Paragraph 1: Please specify the brand or manufacturer of the kefir grains used in this study. If artisanal cultures were used, kindly provide the source information as well.

Response 7: Thanks for reviewer’s suggestions. We've added sources of kefir grains to the manuscript on page 4 line 2.

Comments 8: Section 2.1, Paragraph 2: I recommend changing the fermentation temperature into degree Celsius.

Response 8: Thanks for reviewer’s suggestions. We've revised it in the text.

Comments 9: Section 2.1, Paragraph 3: How can the authors ensure that the micro-fermentation (in-vial) performed in this study was reliable, applicable, and produced results comparable to traditional kefir fermentation?

Response 9: Thanks for reviewer’s question. The environment, temperature and other conditions we use for testing were the same as these in a regular kefir fermentation. The ratio of milk to kefir grains used is also the same. In addition, we have measured NMR spectra after fermentation under the same conditions in a normal vessel and the result obtained are comparable to that in situ monitoring.

Comments 10: Figure 3, 4 and 5: It would be beneficial for the authors to present a combined figure with overlaid spectra, highlighting specific regions to demonstrate the effectiveness of PSYCHE and GEMSTONE techniques in enhancing resolution, improving signal extraction, and enabling better quantification compared to traditional 1H-NMR spectra.

Response 10: Thank you for the constructive comments. We have adjusted the figures by merging Figures 4 and 5 to allow for a better comparison of the milk spectra before and after fermentation. Additionally, we have added an enlarged view of the 1D 1H and PSYCHE spectra at the lactose region in Figure 4.

Comments 11: Results and discussion: In the last paragraph, the potential applications and benefits of the improved NMR techniques by PSYCHE and GEMSTONE investigated in this study, particularly in relation to the QC procedures in industrial fermented milk manufacturing, should be suggested.

Response 11: Thank you for the constructive comments. We have added the advantages of the PSYCHE and GEMSTONE methods for QC procedures in industrial fermented milk production to the last paragraph of the Results and discussion.

Comments 12: Conclusions: The conclusion should be re-checked again after considering the collective comments above.

Response 11: Thanks for reviewer’s suggestions.

4. Response to Comments on the Quality of English Language

Point 1: The English is fine and does not require any improvement.  

Round 2

Reviewer 2 Report

Comments and Suggestions for Authors

Thank you for considering my comments. The authors have provided a sufficient effort to improve the quality of manuscript.